# Learning Chordal Markov Networks
# via Branch and Bound

**Kari Rantanen**
HIIT, Dept. Comp. Sci.,
University of Helsinki

**Antti Hyttinen**
HIIT, Dept. Comp. Sci.,
University of Helsinki

**Matti Järvisalo**
HIIT, Dept. Comp. Sci.,
University of Helsinki

## Abstract

We present a new algorithmic approach for the task of finding a chordal Markov network structure that maximizes a given scoring function. The algorithm is based on branch and bound and integrates dynamic programming for both domain pruning and for obtaining strong bounds for search-space pruning. Empirically, we show that the approach dominates in terms of running times a recent integer programming approach (and thereby also a recent constraint optimization approach) for the problem. Furthermore, our algorithm scales at times further with respect to the number of variables than a state-of-the-art dynamic programming algorithm for the problem, with the potential of reaching 20 variables and at the same time circumventing the tight exponential lower bounds on memory consumption of the pure dynamic programming approach.

## 1   Introduction

Graphical models offer a versatile and theoretically solid framework for various data analysis tasks [1, 30, 17]. In this paper we focus on the structure learning task for *chordal Markov networks* (or chordal/triangulated Markov random fields or decomposable graphs), a central class of undirected graphical models [7, 31, 18, 17]. This problem, chordal Markov network structure learning (CMSL), is computationally notoriously challenging; e.g., finding a maximum likelihood chordal Markov network with bounded structure complexity (clique size) is known to be NP-hard [23]. Several Markov chain Monte Carlo (MCMC) approaches have been proposed for this task in the literature [19, 27, 10, 11].

Here we take on the challenge of developing a new *exact* algorithmic approach for finding an *optimal* chordal Markov network structure in the score-based setting. Underlining the difficulty of this challenge, first exact algorithms for CMSL have only recently been proposed [6, 12, 13, 14], and generally do no scale up to 20 variables. Specifically, the constraint optimization approach introduced in [6] does not scale up to 10 variables within hours. A similar approach was also taken in [16] in the form of a direct integer programming encoding for CMSL, but was not empirically evaluated in an exact setting. Comparably better performance, scaling up to 10 (at most 15) variables, is exhibited by the integer programming approach implemented in the GOBNILP system [2], extending the core approach of GOBNILP to CMSL by enforcing additional constraints. The true state-of-the-art exact algorithm for CMSL, especially when the clique size of the networks to be learned is not restricted, is Junctor, implementing a dynamic programming approach [13]. The method is based on recursive characterization of clique trees and storing in memory the scores of already-solved subproblems. Due to its nature, the algorithm has to iterate through every single solution candidate, although its effective memoization technique helps to avoid revisiting solution candidates [13]. As typical for dynamic programming algorithms, the worst-case and best-case performance coincide: Junctor is guaranteed to use $\Omega(4^n)$ time and $\Omega(3^n)$ space.

In this work, we develop an alternative exact algorithm for CMSL. While a number of branch-and-bound algorithms have been proposed in the past for Bayesian network structure learning

(BNSL) [25, 28, 20, 29, 26], to the best of our knowledge our approach constitutes the first non-trivial branch-and-bound approach for CMSL. Our core search routine takes advantage of similar ideas as a recently proposed approach for optimally solving BNSL [29], and, on the other hand, like GOBNILP, uses the tight connection between BNSL and CMSL by searching over the space of chordal Markov network structures via considering decomposable directed acyclic graphs. Central to the efficiency of our approach is the integration of dynamic programming over Bayesian network structures for obtaining strong bounds for effectively pruning the search space during search, as well as problem-specific dynamic programming for efficiently implementing domain filtering during search. Furthermore, we establish a condition which enables symmetry breaking for noticeably pruning the search space over which we perform branch and bound. In comparison with Junctor, a key benefit of our approach is the potential of avoiding worst-case behavior, especially in terms of memory usage, based on using strong bounds to rule out provably non-optimal solutions from consideration during search.

Empirically, we show the approach dominates the integer programming approach of GOBNILP [2], and thereby also the constraint optimization approach [6, 12]. Furthermore, our algorithm scales at times further in terms of the number of variables than the DP-based approach implemented in Junctor [13], with the potential of reaching 20 variables within hours and at the same time circumventing the tight exponential lower bounds on memory consumption of the pure dynamic programming approach, which is witnessed also in practice by noticeably lower memory consumption.[1]

## 2 Chordal Markov Network Structure Learning

A Markov network structure is represented by an undirected graph $G^u = (V, E^u)$, where $V = \{v_1, \ldots, v_n\}$ is the set of vertices and $E^u$ the set of undirected edges. This structure represents independencies $v_i \perp\!\!\!\perp v_j | S$ according to the undirected separation property: $v_i$ and $v_j$ are separated given set $S$ if and only if all paths between them go through a vertex in set $S$. The undirected graph is *chordal* iff every (undirected) cycle of length greater than three contains a *chord*, i.e., an edge between two non-consecutive vertices in the cycle. Figure 1 a) shows an example. Here we focus on the task of finding a chordal graph $U$ that maximizes posterior probability $P(G^u|D) = P(D|G^u)P(G^u)/P(D)$, where $D$ denotes the i.i.d. data set. As we assume a uniform prior over chordal graphs, this boils down to maximizing the marginal likelihood $P(D|G^u)$.

Dawid et al. have shown that the marginal likelihood $P(D|G^u)$ for chordal Markov networks can be calculated using a clique tree representation [7, 9]. A clique $C$ is a fully connected subset of vertices. A clique tree for an undirected graph $G^u$ is an undirected tree such that

   I.  $\bigcup_i C_i = V$,
   II.  if $\{v_\ell, v_k\} \in E^u$, then either $\{v_\ell, v_k\} \subseteq C_k$ or $\{v_\ell, v_k\} \subseteq C_\ell$, and
   III.  the running intersection property holds: whenever $v_k \in C_i$ and $v_k \in C_j$, then $v_k$ is also in every clique on the unique path between $C_i$ and $C_j$.

The separators are the intersections of adjacent cliques in a clique tree. Figure 1 b) shows an example. The marginal likelihood factorizes according to the clique tree: $P(D|U) = \prod_i P(C_i)/\prod_j P(S_j)$ (assuming positivity and that the prior factorizes) [6]. The marginal likelihood $P(S)$ for a set $S$ of random variables can be calculated with suitable priors; in this paper we consider discrete data using a Dirichlet prior. If we denote $s(S) = \log P(S)$, CMSL can be cast as maximizing $\sum_{C_i} s(C_i) - \sum_{S_j} s(S_j)$. For example, the marginal log-likelihood of the graph in Figure 1 a) can be calculated using the clique tree presentation in Figure 1 b) as $s(\{v_1, v_6\}) + s(\{v_1, v_5\}) + s(\{v_1, v_2, v_3\}) + s(\{v_2, v_3, v_4\}) - s(\{v_1\}) - s(\{v_1\}) - s(\{v_2, v_3\})$.

In this paper, we view the chordal Markov network structure learning problem from the viewpoint of directed graphs, making use of the fact that for each chordal Markov network structure there are equivalent directed graph structures [15, 7], which we call here decomposable DAGs. A decomposable DAG is a DAG $G = (V, E)$ such that the set of directed edges $E \subset V \times V$ does not include any *immoralities*, i.e., structures of the form $v_i \rightarrow v_k \leftarrow v_j$ with no edges between $v_i$ and $v_j$. Due to lack of immoralities, the d-separation property on a decomposable DAG corresponds exactly to the separation property on the chordal undirected graph (the skeleton of the decomposable DAG). Thus, decomposable graphs represent distributions that are representable by Markov and by

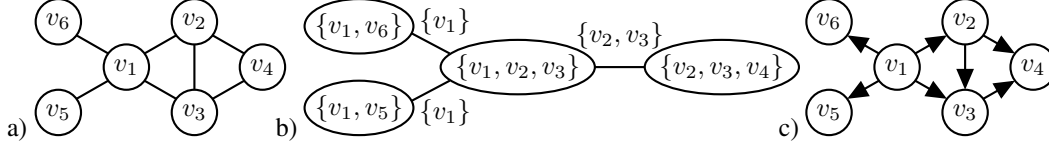

Figure 1: Three views on chordal Markov network structures: a) chordal undirected graph, b) clique tree, (c) decomposable DAG.

Bayesian networks. Figure 1 c) shows a corresponding decomposable DAG for the chordal undirected graph in a). Note that the decomposable DAG may not be unique; for example, $v_2 \rightarrow v_3$ can be directed also in the opposite direction. The score of the decomposable DAG can be calculated as $s(v_1, \emptyset) + s(v_5, \{v_1\}) + s(v_6, \{v_1\}) + s(v_2, \{v_1\}) + s(v_3, \{v_1, v_2\}) + s(v_4, \{v_2, v_3\})$, where $s(v_i, S)$ are the local scores for BNSL using e.g. a Dirichlet prior. Because these local scores $s(\cdot, \cdot)$ correspond to $s(\cdot)$ through $s(v_i, S) = s(\{v_i, S\}) - s(S)$ (and $s(\emptyset) = 0$), we find that this BNSL scoring gives the same result as the clique tree based scoring rule.

Thus CMSL can also be cast as the optimization problem of finding a graph in

$$\arg\max_{G \in \mathcal{G}} \sum_{v_i \in V} s(v_i, \mathrm{pa}_G(v_i)),$$

where $\mathcal{G}$ denotes the class of decomposable DAGs. (This formulation is used also in the GOBNILP system [2].) The optimal chordal Markov network structure is the skeleton of the optimal $G$. This problem is notoriously computationally difficult in practice, emphasized by the fact that standard score-pruning [3, 8] used for BNSL is not generally applicable in the context of CMSL as it will often prevent finding the true optimum: pruning parent sets for vertices often circumvents other vertices achieving high scoring parents sets (as immoralities would be induced).

## 3 Hybrid Branch and Bound for CMSL

In this section we present details on our branch-and-bound approach to CMSL. We start with an overview of the search algorithm, and then detail how we apply symmetry breaking and make use of dynamic programming to dynamically update variable domains, i.e., for computing parent set choices during search, and to obtain tight bounds for pruning the search tree.

### 3.1 Branch and Bound over Ordered Decomposable DAGs

The search is performed over the space of *ordered decomposable DAGs*. While in general the order of the vertices of a DAG can be ambiguous, this notion allows for differentiating the exact order of the vertices, and allows for pruning the search space by identifying symmetries (see Section 3.2).

**Definition 1.** $G = (V, E, \pi)$ *is an ordered decomposable DAG if and only if* $(V, E)$ *is a decomposable DAG and* $\pi : \{1...n\} \rightarrow \{1...n\}$ *a total order over $V$ such that* $(v_i, v_j) \in E$ *only if* $\pi^{-1}(i) < \pi^{-1}(j)$ *for all* $v_i, v_j \in V$.

Partial solutions during search are hence ordered decomposable DAGs, which are extended by adding a parent set choice $(v, P)$, i.e., adding the new vertex $v$ and edges from each of its parents in $P$ to $v$.

**Definition 2.** *Let $G = (V, E, \pi)$ be an ordered decomposable DAG. Given $v_k \notin V$ and $P \subseteq V$, we say that the ordered decomposable DAG $G' = (V', E', \pi')$ is $G$ with the parent set choice $(v_k, P)$ if the following conditions hold.*

1. $V' = V \cup \{v_k\}$
2. $E' = E \cup \bigcup_{v' \in P} \{(v', v_k)\}$.
3. *We have* $\pi'(i) = \pi(i)$ *for all* $i = 1...|V|$, *and* $\pi'(|V| + 1) = k$.

Algorithm 1 represents the core functionality of the branch and bound. The recursive function takes two arguments: the remaining vertices of the problem instance, $U$, and the current partial solution $G = (V, E, \pi)$. In addition we keep stored a best lower bound solution $G^*$, which is the

---
**Algorithm 1** The core branch-and-bound search.
---
1: **function** BRANCHANDBOUND($U, G = (V, E, \pi)$)
2:     **if** $U = \emptyset$ **and** $s(G^*) < s(G)$ **then** $G^* \leftarrow G$             $\triangleright$ Update LB if improved.
3:     **if** this branch cannot improve LB **then return**                $\triangleright$ Backtrack
4:     **for** $(v_i, P) \in$ PARENTSETCHOICES($U, G$) **do**    $\triangleright$ Iterate the current parent set choices.
5:         Let $G' = (V', E', \pi')$ be $G$ with the parent set choice $(v_i, P)$.
6:         BRANCHANDBOUND($U \setminus \{v_i\}, G'$)                $\triangleright$ Continue the search.
---

highest-scoring solution that has been found so far. Thus, at the end of the search, $G^*$ is an optimal solution. During the search we use $G^*$ for bounding as further detailed in Section 3.3.

In the loop on line 4 we branch with all the parent set choices that we have deemed necessary to try during the search. The method PARENTSETCHOICES($U, G$) and the related symmetry breaking are explained in Section 3.2. We sort the parent set choices into decreasing order based on their score, so that $(v, P)$ is tried before $(v', P')$ if $s(v, P) > s(v', P')$, where $v, v' \in U$ and $P, P' \subseteq V$. This is done to focus the search first to the most promising branches for finding an optimal solution. When $U = \emptyset$, we have PARENTSETCHOICES($U, G$) = $\emptyset$, and so the current branch gets terminated.

## 3.2 Dynamic Branch Selection, Parent Set Pruning, and Symmetry Breaking

We continue by proposing symmetry breaking for the space of ordered decomposable DAGs, and propose a dynamic programming approach for dynamic parent set pruning during search. We start with symmetry breaking.

In terms of our branch-and-bound approach to CMSL, symmetry breaking is a vital part of the search, as there can be exponentially many decomposable DAGs which correspond to a single undirected chordal graph; for example, the edges of a complete graph can be directed arbitrarily without the resulting DAG containing any immoralities. Hence symmetry breaking in terms of pruning out symmetric solution candidates during search has potential for noticeably speeding up search.

Chickering [4, 5] showed how so-called covered edges can be used to detect equivalencies between Bayesian network structures. Later van Beek and Hoffmann [29] implemented covered edge based symmetry breaking in their BNSL approach. Here we introduce the concept of *preferred vertex orders*, which generalizes covered edges for CMSL based on the decomposability of the solution graphs.

**Definition 3.** *Let $G = (V, E, \pi)$ be an ordered decomposable DAG. A pair $v_i, v_j \in V$ violates the preferred vertex order in $G$ if the following conditions hold.*

    *1. $i > j$.*
    *2. $pa_G(v_i) \subseteq pa_G(v_j)$.*
    *3. There is a path from $v_i$ to $v_j$ in $G$.*

Theorem 1 states that for any (partial) solution (i.e., an ordered decomposable DAG), there always exists an equivalent solution that does not contain any violations of the preferred vertex order. Mapping to practice, this theorem allows for very effectively pruning out all symmetric solutions but the one not violating the preferred vertex order within our branch-and-bound approach. A detailed proof is provided in Appendix A.

**Theorem 1.** *Let $G = (V, E, \pi)$ be an ordered decomposable DAG. There exists an ordered decomposable DAG $G' = (V, E', \pi')$ that is equivalent to $G$, but where for all $v_i, v_j \in V$ the pair $(v_i, v_j)$ does not violate the preferred vertex order in $G'$.*

It follows from Theorem 1 that for each solution (ordered decomposable DAG) there exists an equivalent solution where the lexicographically smallest vertex is a source. Thus we can fix it as the first vertex in the order at the beginning of the search.

Similarly as in [29] for BNSL, we define the depths of vertices as follows.

**Definition 4.** *Let $G = (V, E, \pi)$ be an ordered decomposable DAG. The depth of $v \in V$ in $G$ is*

$$d(G, v) = \begin{cases} 0 & \text{if } pa_G(v) = \emptyset, \\ \max_{v' \in pa_G(v)} d(G, v') + 1 & \text{otherwise.} \end{cases}$$

*The depths of $G$ are* ordered *if for all $v_i, v_j \in V$, where $\pi^{-1}(i) < \pi^{-1}(j)$, the following hold. 1. $d(G, v_i) \leq d(G, v_j)$, and 2. If $d(G, v_i) = d(G, v_j)$, then $i < j$.*

Note that "violating the preferred vertex order" concerns the order in which the vertices are in the underlying DAG, whereas "depths are ordered" concerns the order by which a solution was constructed. We use the former to prune whole solution candidates from the search space, and the latter to ensure that no solution candidate is seen twice during search.

We also propose a dynamic programming approach to branch selection and parent set pruning during search, based on the following definition of valid parent sets.

**Definition 5.** *Let $G = (V, E, \pi)$ be an ordered decomposable DAG. Given $v_k \notin V$ and $P \subseteq V$, let $G' = (V', E', \pi')$ be $G$ with the parent set choice $(v_k, P)$. The parent set choice $(v_k, P)$ is* valid *for $G$ if the following hold.*

> 1. *For all $v_i, v_j \in P$ we have either $(v_i, v_j) \in E$ or $(v_j, v_i) \in E$.*
> 2. *For all $v_i \in V$, the pair $(v_i, v_k)$ does not violate the preferred vertex order in $G'$.*
> 3. *The depths of $G'$ are ordered.*

Given a partial solution $G = (V, E, \pi)$, a vertex $v \notin V$, and a subset $P \subseteq V$, the function GETSU-PERSETS in Algorithm 2 represents a dynamic programming method for determining valid parent set choices $(v, P')$ for $G$ where $P' \supseteq P$. An advantage of this formulation is that invalidating conditions for a parent set, such as immoralities or violations of the preferred vertex order, automatically hold for all the supersets of the parent set; this is applied on line 6 to avoid unnecessary branching.

On line 8 we require that a parent set $P$ is added to the list only if none of its valid supersets $P' \in \mathcal{C}$ have a higher score. This pruning technique is based on the observation that $P'$ provides all the same moralizing edges as $P$, and therefore it is sufficient to only consider the parent set choice $(v, P')$ in the search when $s(v, P) \leq s(v, P')$.

Given the set of remaining vertices $U$, the function PARENTSETCHOICES in Algorithm 2 constructs all the available parent set choices for the current partial solution $G = (V, E, \pi)$. The collection $\mathcal{M}(G, v_i)$ contains the *subset-minimal parent sets* for vertex $v_i \in U$ that satisfy the 3rd condition of Definition 5. If $V = \emptyset$, then $\mathcal{M}(G, v_i) = \{\emptyset\}$. Otherwise, let $k$ be the maximum depth of the vertices in $G$. Now $\mathcal{M}(G, v_i)$ contains the subset-minimal parent sets that would insert $v_i$ on depth $k + 1$. In addition, if $i > j$ for all $v_j \in V$ where $d(G, v_j) = k$, then $\mathcal{M}(G, v_i)$ also contains the subset-minimal parent sets that would insert $v_i$ on depth $k$. Note that the cardinality of any parent set in $\mathcal{M}(G, v_i)$ is at most one.

## 3.3 Computing Tight Bounds by Harnessing Dynamic Programming for BNSL

To obtain tight bounds during search, we make use of the fact that the score of the optimal BN structures for the BNSL instance with same scores as in the CMSL instance at hand is guaranteed to give an upper bound on the optimal solutions to the CMSL instance. To compute an optimal BN structure, we use a variant of a standard dynamic programming algorithm by Silander and Myllymäki [22]. While there are far more efficient algorithms for BNSL [2, 32, 29], we use BNSL DP for obtaining an upper bound during the branch-and-bound search under the current partial

---

**Algorithm 2** Constructing parent set choices via dynamic programming.

1: **function** PARENTSETCHOICES($U, G = (V, E, \pi)$)
2:      **return** $\bigcup\limits_{v \in U} \bigcup\limits_{M \in \mathcal{M}(G,v)}$ GETSUPERSETS($v, G, M$)

3: **function** GETSUPERSETS($v, G = (V, E, \pi), P$)
4:      Let $\mathcal{C} = \emptyset$
5:      **for** $v' \in V \setminus P \setminus \{v\}$ **do**
6:          **if** $(v, P')$ is a valid parent set choice for $G$ with some $P' \supseteq P \cup \{v'\}$ **then**
7:              $\mathcal{C} \leftarrow \mathcal{C} \cup$ GETSUPERSETS($v, G, P \cup \{v'\}$)
8:      **if** $(v, P)$ is valid parent set choice for $G$ **and** $s(v, P) > s(v, P')$ for all $P' \in \mathcal{C}$ **then**
9:          $\mathcal{C} \leftarrow \mathcal{C} \cup \{(v, P)\}$
10:      **return** $\mathcal{C}$

---

CMSL solution (i.e., under the current branch). Specifically, before the actual branch and bound, we precompute a DP table which stores, for each subset of vertices $V' \subset V$ of the problem instance, the score of the so-called *BN extensions* of $V'$, i.e., the optimal BN structures over $U = V \setminus V'$ where we additionally allow the vertices in $U$ to also take parents from $V'$. This guarantees that the BN extensions are compatible with the vertex order in the current branch of the branch-and-bound search tree, and thereby the sum of the score of the current partial CMSL solution over $V'$ and the score of the optimal BN extensions of $V'$ is a valid upper bound. By spending $\mathcal{O}(n \cdot 2^n)$ time in the beginning of the branch and bound for computing the scores of optimal BN extensions of every $V' \subset V$, we can then look up these scores during branch and bound in $\mathcal{O}(1)$ time.

With the DP table, it takes only low polynomial time to construct the optimal BN structure over the set of all vertices [22], i.e., a BN extension of $\emptyset$. Thus, we can obtain an initial lower bound solution $G^*$ for the branch and bound as follows.

1. Construct the optimal BN structure for the vertices of the problem instance
2. Try to make the BN decomposable by heuristically adding or removing edges.
3. Let $G^*$ be the highest-scoring decomposable DAG from step 2.

However, the upper bounds obtained via BNSL can be at times can be quite weak when the network structures contain many immoralities. For this reason, in Algorithm 3, we introduce an additional method for computing the upper bounds, taking immoralities "relaxedly" into consideration. The algorithm takes four inputs: A fixed partial solution $G = (V, E, \pi)$, a list of vertices $A$ that we have assigned during the upper bound computation, a list of remaining vertices $U$, and an integer $d \geq 0$ which dictates the maximum recursion depth. As a fallback option, on line 3 we return the optimal BN score for the remaining vertices if the maximum recursion depth is reached.

On line 4 we construct the collection of sets $\mathcal{P}$ that are the maximal sets that any vertex can take as parent set during the upper bound computation. The sets in $\mathcal{P}$ take immoralities relaxedly into consideration: For any $v_i, v_j \in V$, we have $\{v_i, v_j\} \subseteq P$ for some $P \in \mathcal{P}$ if and only if $(v_i, v_j) \in E$ or $(v_j, v_i) \in E$. That is, when choosing parent sets during the upper bound computation, we allow immoralities to appear, as long as they are not between vertices of the fixed partial solution. In the loop on line 6, we iterate through each vertex $v \in U$ that is still remaining, and find its highest-scoring relaxedly-moral parent set according to $\mathcal{P}$. Note that given any $P' \in \mathcal{P}$, we can find the highest-scoring parent set $P \subseteq P'$ in $\mathcal{O}(1)$ time when the scores are stored in a segment tree. For information about constructing such data structure, see [22]. Thus line 7 takes $\mathcal{O}(|V|)$ time to execute. Finally, on line 8 of the loop, we split the problem into subproblems to see which parent set choice $(v, P)$ provides the highest local upper bound $u$ to be returned.

Algorithm 3 requires $\mathcal{O}((n - m) \cdot m \cdot 2^{n-m})$ time, where $m = |V|$ is the number of vertices in the partial solution and $n$ the number of vertices in the problem instance, assuming that the BN extensions and the segment trees have been precomputed. (In the empirical evaluation, the total runtimes of our branch-and-bound approach include these computations.) The collections $\mathcal{P}$ can exist implicitly.

We use the upper bounds within branch and bound as follows. Let $G = (V, E, \pi)$ be the current partial solution, let $U$ be the set of remaining vertices, and let $b$ be the score of optimal BN extensions of $V$. We can close the current branch if $s(G^*) \geq s(G) + b$. Otherwise, we can close the branch if $s(G^*) \geq s(G) + \text{UPPERBOUND}(G, \emptyset, U, d)$ for some $d > 0$. Our implementation uses $d = 10$.

---

**Algorithm 3** Computing upper bounds for a partial solution via dynamic programming.

---
1: **function** UPPERBOUND($G = (V, E, \pi), A, U, d$)
2:     **if** $U = \emptyset$ **then return** 0
3:     **if** $d = 0$ **then return** the score of optimal BN extensions of $V \cup A$
4:     Let $\mathcal{P} = \bigcup_{v \in V} \{\{v\} \cup pa_G(v) \cup A\}$
5:     Let $u \leftarrow -\infty$
6:     **for** $v \in U$ **do**
7:         Let $P = \underset{P \subseteq P' \in \mathcal{P}}{\arg\max}\, s(v, P)$
8:         $u \leftarrow \max(u, s(v, P) + \text{UPPERBOUND}(G, A \cup \{v\}, U \setminus \{v\}, d - 1))$
9:     **return** $u$

---

# 4 Empirical Evaluation

We implemented the branch-and-bound algorithm in C++, and refer to this prototype as BBMarkov. We compare the performance of BBMarkov to that of GOBNILP (the newest development version [24] at the time of publication, using IBM CPLEX version 12.7.1 as the internal IP solver) as a state-of-the-art BNSL system implementing a integer programming branch-and-cut approach to CMSL by ruling out non-chordal graphs, and Junctor, implementing a state-of-the-art DP approach to CMSL. We used a total of 54 real-world datasets used as standard benchmarks for exact approaches [32, 29]. For investigating scalability of the algorithms in terms of the number of variables $n$, we obtained from each dataset several benchmark instances by restricting to the first $n$ variables for increasing values of $n$. We did not impose a bound on the treewidth of the chordal graphs of interest, i.e., the size of candidate parent sets was not limited. We used the BDeu score with equivalent sample size 1. As standard practice in benchmarking exact structure learning algorithms, we focus on comparing the running times of the considered approaches on precomputed input CMSL instances. The experiments were run under Debian GNU/Linux on 2.83-GHz Intel Xeon E5440 nodes with 32-GB RAM.

Figure 2 compares BBMarkov to GOBNILP and Junctor under a 1-h per-instance time limit, with different numbers $n$ of variables distinguished using different point styles. BBMarkov clearly dominates GOBNILP in runtime performance (Fig. 2 left); instances for $n > 15$ are not shown as GOBNILP was unable to solve them. Compared to Junctor (Fig. 2 middle, Table 1), BBMarkov exhibits complementary performance. Junctor is noticeably strong on several datasets and lower values of $n$, and exhibits fewer timeouts. For a fixed $n$, Junctor's runtimes have a very low variance independent of the dataset, which is due to the $\Omega(4^n)$ (both worst-case and best-case) runtime guarantee. However, BBMarkov shows potential for scaling up for larger $n$ than Junctor: at $n = 17$ Junctor's runtimes are very close to 1 h on all instances, while BBMarkov's bounds rule out at times very effectively non-optimal solutions, resulting in noticeable lower runtimes on specific datasets with increasing $n$. This is show-cased in Table 1 on the right, highlighting some of the best-case performance of BBMarkov using per-instance time limit of 24 h for both BBMarkov and Junctor.

In terms of how the various search techniques implemented in BBMarkov contribute to the running times of BBMarkov, we observed that the running times for obtaining BNSL-based bounds (via the use of exact BN dynamic programming and segment trees) tend to be only a small fraction of the overall running times. For example, at $n = 20$, these computations take less than minute in total. Most of the time in the search is typically used in the optimization loop and in computing the tighter upper bounds that take immoralities "relaxedly" into consideration. While computing the tighter bounds is more expensive than computing the exact BNs at the beginning of search, the tighter bounds often pay off in terms of overall running times as branches can be closed earlier during search.

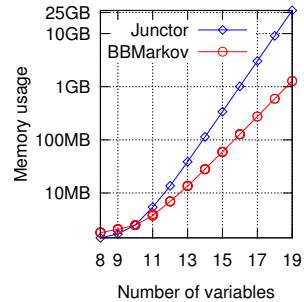

Figure 3: Memory usage

Another benefit of BBMarkov compared to Junctor is the observed lower memory consumption (Figure 3). Junctor's $\Omega(3^n)$ memory

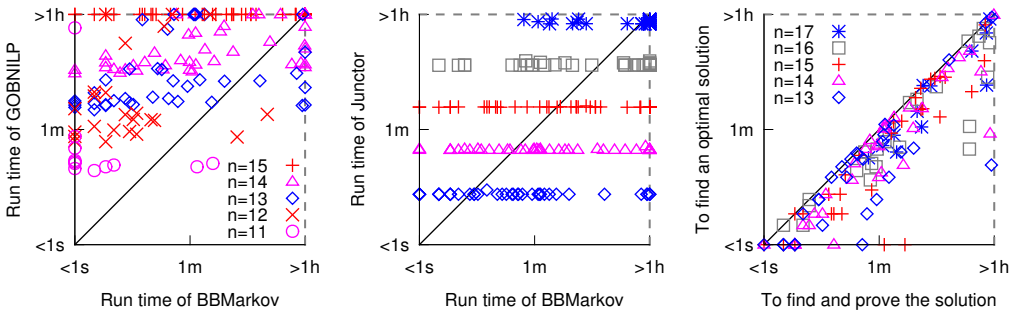

Figure 2: Per-instance runtime comparisons. Left: BBMarkov vs GOBNILP. Middle: BBMarkov vs Junctor. Right: BBMarkov time to finding vs BBMarkov time to proving an optimal solution.

Table 1: BBMarkov v Junctor. Left: smaller datasets and for different sample sizes on the Water dataset. Right: Examples of best-case performance of BBMarkov. **to**: timeout, **mo**: memout.

| Dataset | $n$ | BBMarkov | | Junctor |
|---|---|---|---|---|
| Wine | 13 | <1 | (<1) | 6 |
| Adult | 14 | 58 | (35) | 29 |
| Letter | 16 | >3600 | (>3600) | 592 |
| Voting | 17 | 281 | (207) | 3050 |
| Zoo | 17 | >3600 | (>3600) | 2690 |
| Water100 | 17 | 100 | (49) | 2580 |
| Water1000 | 17 | 2731 | (279) | 2592 |
| Water10000 | 17 | >3600 | (>3600) | 2928 |
| Tumor | 18 | 610 | (268) | 12019 |

| Dataset | $n$ | BBMarkov | | Junctor |
|---|---|---|---|---|
| Alarm | 17 | 268 | (62) | 2724 |
| | 18 | 1462 | (315) | 12477 |
| | 19 | 10274 | (2028) | 52130 |
| | 20 | 49610 | (50) | **mo** |
| Heart | 17 | 41 | (22) | 3007 |
| | 18 | 162 | (85) | 11179 |
| | 19 | 1186 | (698) | 50296 |
| | 20 | 15501 | (13845) | **mo** |
| Hailfinder500 | 17 | 225 | (108) | 2588 |
| | 18 | 2543 | (1348) | 12422 |
| | 19 | 13749 | (6418) | 53108 |
| | 20 | 33503 | (25393) | **mo** |
| Water100 | 18 | 590 | (244) | 12244 |
| | 19 | 6581 | (6187) | 52575 |
| | 20 | 61152 | (54806) | **mo** |

usage results consistently in running out on memory for $n \geq 20$. At $n = 19$, BBMarkov uses on average approx. 1 GB of memory, while Junctor uses close to 30 GB. A further benefit of BBMarkov is its ability to provide "anytime" solutions during search. In fact, the bounds obtained during search result at times in finding optimal solutions relatively fast: Figure 2 right shows the ratio of time needed to find an optimal solution (x-axis) from time needed to terminate search, i.e., to find a solution and prove its optimality (y-axis), and in Table 1, with the time needed to find an optimal solution given in parentheses.

## 5    Conclusions

We introduced a new branch-and-bound approach to learning optimal chordal Markov network structures, i.e., decomposable graphs. In addition to core branch-and-bound search, the approach integrates dynamic programming for obtaining tight bounds and effective variable domain pruning during search. In terms of practical performance, the approach has the potential of reaching 20 variables within hours of runtime, at which point the competing native dynamic programming approach Junctor runs out of memory on standard modern computers. When approaching 20 variables, our approach is approximately 30 times as memory-efficient as Junctor. Furthermore, in contrast to Junctor, the approach is "anytime" as solutions can be obtained already before finishing search. Efficient parallelization of the approach is a promising direction for future work.

### Acknowledgments

The authors gratefully acknowledge financial support from the Academy of Finland under grants 251170 COIN Centre of Excellence in Computational Inference Research, 276412, 284591, 295673, and 312662; and the Research Funds of the University of Helsinki.

## A    Proofs

We give a proof for Theorem 1, central in enabling effective symmetric breaking in our branch-and-bound approach. We start with a definition and lemma towards the proof.

**Definition 6.** *Let $V = \{v_1, ..., v_n\}$ be a set of vertices and let $\pi$ and $\pi'$ be some total orders over $V$. Let $k = \min_{i, \pi(i) \neq \pi'(i)} i$ be the first difference between the orders. If no such difference exists, we denote $\pi = \pi'$. Otherwise we denote $\pi < \pi'$ if and only if $\pi(k) < \pi'(k)$.*

**Lemma 1.** *Let $G = (V, E, \pi)$ be an ordered decomposable DAG. If there are $v_i, v_j \in V$ such that the pair $(v_i, v_j)$ violates the preferred vertex order in $G$, then there exists an ordered decomposable DAG $G' = (V, E', \pi')$, where 1. $G'$ belongs to the same equivalence class with $G$, 2. the pair $(v_i, v_j)$ does not violate the preferred vertex order in $G'$, and 3. $\pi < \pi'$.*

*Proof.* We begin by defining a directed clique tree $\mathcal{C} = (\mathcal{V}, \mathcal{E})$ over $G$.

Given $v_k \in V$, let $C_k = pa_G(v_k) \cup \{v_k\}$ be the clique defined by $v_k$ in $G$. The vertices of $\mathcal{C}$ are these cliques; we also add an empty set as a clique to make sure the cliques form a tree (and not a forest). Formally, $\mathcal{V} = \{C_k \mid v_k \in V\} \cup \{\emptyset\}$.

Given $v_k \in V$, where $pa_G(v_k) \neq \emptyset$, let $\phi_k = \operatorname{argmax}_{v_\ell \in pa_G(v_k)} \pi^{-1}(\ell)$ denote the parent of $v_k$ in $G$ that is in the least significant position in $\pi$. Now, the edges of $\mathcal{C}$ are

$$\mathcal{E} = \{(\emptyset, C_k) \mid C_k = \{v_k\}, v_k \in V\} \cup \{(C_\ell, C_k) \mid v_\ell = \phi_k, C_k \neq \{v_k\}, v_k \in V\}.$$

In words, if $v_k \in V$ is a source vertex in $G$ (i.e., $C_k = \{v_k\}$), then the parent of $C_k$ is $\emptyset$ in $\mathcal{C}$. Otherwise (i.e., $C_k \neq \{v_k\}$) the parent of $C_k$ is $C_\ell$, where $v_\ell$ is the closest vertex to $v_k$ in order $\pi$ that satisfies $C_\ell \cap pa_G(v_k) \neq \emptyset$. We see that all the requirements for clique trees hold for $\mathcal{C}$: I. $\bigcup_{C \in \mathcal{V}} C = V$, II. if $\{v_\ell, v_k\} \in E$, then either $\{v_\ell, v_k\} \subseteq C_k$ or $\{v_\ell, v_k\} \subseteq C_\ell$, and III. due to the decomposability of $G$, we have $C_a \cap C_c \subseteq C_b$ on any path from $C_a$ to $C_c$ through $C_b$ (the running intersection property).

Now assume that there are $v_i, v_j \in V$ such that the pair $(v_i, v_j)$ violates the preferred vertex order in $G$; that is, we have $i > j$, $pa_G(v_i) \subseteq pa_G(v_j)$ and a path from $v_i$ to $v_j$ in $G$. This means that there is a path from $C_i$ to $C_j$ in $\mathcal{C}$ as well.

Let $P \in \mathcal{V}$ be the parent vertex of $C_i$ in $\mathcal{C}$. We see that $C_j$ exists in a subtree $\mathcal{T}$ of $\mathcal{C}$ that is separated from rest of $\mathcal{C}$ by $P$, and where $C_i$ is the root vertex. Let $\mathcal{T}'$ be a new clique tree that is like $\mathcal{T}$, but redirected so that $C_j$ is the root vertex of $\mathcal{T}'$. Let $\mathcal{C}'$ be a new clique tree that is like $\mathcal{C}$, but $\mathcal{T}$ is replaced with $\mathcal{T}'$.

We show that $\mathcal{C}'$ is a valid clique tree. First of all, the vertices (cliques) of $\mathcal{C}'$ are exactly the same as in $\mathcal{C}$, so $\mathcal{C}'$ clearly satisfies the requirements I and II. As for the requirement III, consider the non-trivial case where $C_a, C_b \in \mathcal{C}$ have a path from $C_a$ to $C_b$ through $C_i$ in $\mathcal{C}$. This means $v_i \notin C_a$ (due to the way $\mathcal{C}$ was constructed), and so we get

$$C_a \cap C_b \subseteq C_i \quad \rightarrow C_a \cap C_b \subseteq C_i \setminus \{v_i\} \quad \rightarrow C_a \cap C_b \subseteq pa_G(v_i) \underset{\text{Def. 3 (2)}}{\subseteq} pa_G(v_j) \subseteq C_j.$$

Therefore the running intersection property holds for $\mathcal{C}'$.

Let $\hat{\pi}$ be the total order by which $\mathcal{C}'$ is ordered. Let $G' = (V, E', \hat{\pi})$ be a new ordered decomposable DAG that is equivalent to $G$, but where the edges $E'$ are arranged to follow the order $\hat{\pi}$.

Finally, we see that $G'$ satisfies the conditions of the theorem: 1. The cliques of $G'$ are identical to that of $G$, so $G'$ belongs to the same equivalence class with $G$. 2. We have $\hat{\pi}^{-1}(j) < \hat{\pi}^{-1}(i)$, and therefore there is no path from $v_i$ to $v_j$ in $G'$. Thus the pair $(v_i, v_j)$ does not violate the preferred vertex order in $G'$. 3. Let $o = \pi^{-1}(i)$. We have $\hat{\pi}(o) = j < i = \pi(o)$. Furthermore, the change from $\mathcal{T}$ to $\mathcal{T}'$ in $\mathcal{C}'$ did not affect any vertex whose position was earlier than $o$. Therefore $\hat{\pi}(k) = \pi(k)$ for all $k = 1...(o-1)$. This implies $\hat{\pi} < \pi$. □

*Proof of Theorem 1.* Consider the following procedure for finding $G'$.

1. Select $v_i, v_j \in V$ where the pair $(v_i, v_j)$ violates the preferred vertex order in $G$. If there are no such vertices, assign $G' \leftarrow G$ and terminate.
2. Let $\pi$ be the total order of the vertices of $G$. Construct an ordered decomposable DAG $\hat{G} = (V, \hat{E}, \pi')$ such that I. the pair $(v_i, v_j)$ does not violate the preferred vertex order in $\hat{G}$, II. $\hat{G}$ belongs to the same equivalent class with $G$, and III. $\pi' < \pi$. By Lemma 1, $\hat{G}$ can be constructed from $G$.
3. Assign $G \leftarrow \hat{G}$ and return to step 1.

It is clear that when the procedure terminates, $G'$ belongs to same equivalence class with $G$ and there are no violations of the preferred vertex order in $G'$. We also see that the total order of $G$ (i.e., $\pi$) is lexicographically strictly decreasing every time the step 3 is reached. There are finite amount of possible permutations (total orders) and therefore the procedure converges. The existence of this procedure and its correctness proves that $G'$ exists. □

## Footnotes

[1]Extended discussion and empirical results are available in [21].

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
