[Reviews · NeurIPS 2017]

Reviewer 1



The authors present a branch and bound algorithm for learning Chordal Markov networks. The prior state of the art algorithm is a dynamic programming approach based on a recursive characterization of clique tress and storing in memory the scores of already-solved subproblems. The proposed algorithm uses a branch and bound algorithm to search for an optimal chordal Markov network. The algorithm first uses a dynamic programming algorithm to enumerate Bayesian network structures, which are later used as pruning bounds. A symmetry breaking technique is introduced to prune the search space. The special property of Chordal Markov networks necessitates that no score pruning can be done. Given that this step is exponential in time and space. It is unlikely that learning CMSL can scale to even medium-size datasets (>20-30variables). What is the definition of decomposable DAG? This is important for understanding other concepts defined later on, such as ordered decomposable DAG. The symmetry breaking rule is useful. What is its relation to the symmetry breaking rules introduced in (van Beek&Hoffmann 15) for learning BNs? The authors referenced that paper and should have discussed the relevance. The paper did not give a clear picture on how much time is spent on computing scores, and how much time is spent on optimization. Also, the total runtime of the proposed algorithm includes computing optimal BNs and segment trees. How much do they take relative to the total runtime? The aggregated time does not provide a complete picture.

Reviewer 2



The authors present a set of techniques for improved learning of chordal Markov networks, by combining and refining existing techniques. The paper is very well presented, and the empirical evaluation shows that the proposed techniques display advantages over existing methods. The contributions are somewhat incremental when the state-of-art is considered, but still they offer improvements that may be useful to other researchers. One point about the presentation: the authors do not explain the reasons why "symmetry breaking" is important, and I believe this should be discussed in some detail so as to help the reader understand the main points. It is not clear to me whether the authors use a bound on the number of parents, and how it is used in the experiments. Please add some information about it. Concerning the text, I have a few points to make: - Please order references when they appear together (say at the end of the first paragraph of the Introduction). - Page 1, line -1: "coincide" instead of "coincidence". - Section 2, line 2: $E^u$.